# STREETS: A Novel Camera Network Dataset for Traffic Flow

**Corey Snyder**
University of Illinois
cesnyde2@illinois.edu

**Minh N. Do**
University of Illinois
minhdo@illinois.edu

## Abstract

In this paper, we introduce STREETS, a novel traffic flow dataset from publicly available web cameras in the suburbs of Chicago, IL. We seek to address the limitations of existing datasets in this area. Many such datasets lack a coherent traffic network graph to describe the relationship between sensors. The datasets that do provide a graph depict traffic flow in urban population centers or highway systems and use costly sensors like induction loops. These contexts differ from that of a suburban traffic body. Our dataset provides over 4 million still images across 2.5 months and one hundred web cameras in suburban Lake County, IL. We divide the cameras into two distinct communities described by directed graphs and count vehicles to track traffic statistics. Our goal is to give researchers a benchmark dataset for exploring the capabilities of inexpensive and non-invasive sensors like web cameras to understand complex traffic bodies in communities of any size. We present benchmarking tasks and baseline results for one such task to guide how future work may use our dataset.

## 1 Introduction

Intelligent transportation systems and smart cities have recently become active areas of research with a wide variety of applications. Tasks like pedestrian tracking, traffic forecasting, and vehicle re-identification have attracted interest from researchers in network science, computer vision, graph signal processing, and many other fields. In this paper, we focus on problems associated with vehicular traffic flow in a road network and the data sources that enable this work.

Data for traffic forecasting can come from multiple different perspectives. Third party applications like Google Maps track *floating car data* [1] to extrapolate the traffic state from *probe vehicles*. These probe vehicles are seen as being representative of the current traffic flow. While effective, there are concerns regarding privacy when floating data is used [2]. Furthermore, there is not necessarily effective information sharing with local transportation authorities where it could be beneficial. Automotive manufacturers have begun producing internet-connected vehicles known as *connected vehicles*. These connected vehicles communicate within the Internet of Things at junctions like traffic signals and provide information for understanding traffic, similar to probe vehicles [3]. The distinct advantage here is that the vehicle information can be communicated directly to local authorities through the traffic signals. However, connected vehicle technology is still rather new and concerns about adequate connected vehicle penetration show that it may be a while before enough vehicles are equipped with this technology to leverage this data source [4]. Cities and transit authorities employ a wide array of sensing technology between *in-roadway* and *over-roadway* sensors to monitor traffic. The most popular in-roadway sensor is the inductive loop detector while cameras and radar systems provide over-roadway sensing [5]. In-roadway sensors, while typically accurate and well-controlled, are both expensive and invasive upon installation and maintenance as the road must be physically taken apart. Web cameras are appealing since they are relatively inexpensive, can be deployed noninvasively, and may capture more than just vehicular traffic, e.g. pedestrians and background information such as accidents, construction, and poor road conditions.

We believe that camera networks and their inherent graph-based structure have been under-utilized as a data source for monitoring traffic bodies. In brief, previous work using camera networks has

been limited in both subject matter and context. The relationships between cameras are ignored and most tasks are considered using only a single camera. Furthermore, the settings appear to exclusively be highway systems and urban population centers. The distinct challenges of suburban traffic have largely been ignored. A recent government request for information (RFI) through IARPA identified the same lack of diversity in existing work and a need for new data sources [6].

In this paper, we aim to address these shortcomings by introducing **S**uburban **T**raffic on **GR**aphs using Cam**E**ra N**ET**work**S**, or STREETS[1], a novel traffic flow dataset from publicly available web cameras in the suburbs of Chicago, IL. We collect over 4 million images during 2.5 months across 100 distinct cameras in Lake County, IL. Each camera captures images aperiodically and asynchronously with respect to the other cameras approximately every ten minutes. We partition the cameras into two communities and construct detailed directed graphs to describe the structure of each community. We count vehicles in each image to capture traffic statistics at vertices on these graphs. Lastly, we provide 2,566 documented non-recurring incidents like car accidents and other traffic-interrupting events that coincide with the collected dataset.

It is important to acknowledge the challenges presented by traffic cameras. Varying perspective, lighting conditions, and sources of occlusion all make it difficult to extract meaningful traffic information. In addition, limitations on network bandwidth and data storage force tough design decisions on temporal sampling rate and image resolution. However, we believe that recent advancements in deep learning and computer vision may compensate for these shortcomings. In-roadway sensors may be more reliable, but they are disruptive to deploy and less economical than a camera network and computer vision algorithms. Furthermore, we should not limit our focus to metropolitan areas and highway systems. All types of communities should benefit from research into intelligent transportation systems. Computer vision may be the key to building smarter infrastructure and safer communities at all scales and settings.

The structure of this work is as follows: in Section II, we will elaborate on prior work in greater detail and emphasize the novelty of our dataset. In Section III, we formally introduce STREETS , describe our procedures and algorithms for data collection, and illustrate the content of our dataset with relevant visualizations and metrics. Section IV details traffic forecasting as a possible benchmarking task and corresponding baseline results using STREETS. Finally, we conclude in Section V with a brief discussion of other possible benchmarks using STREETS and plans for improving the dataset. Before we continue, we list the main contributions of this work.

## 1.1 Our Contributions

- STREETS dataset of over 4 million images collected over 2.5 months in two communities in Lake County, IL and 3000 annotated images with 33,260 semantically segmented vehicles.
- Novel presentation of traffic flow data using web cameras with directed graphs.
- 2,566 documented non-recurring traffic-related incidents alongside traffic flow data.
- Benchmark evaluation tasks and baseline results using STREETS.

## 2 Prior Work

### 2.1 Traffic and Vehicle Datasets

The focus of previous vehicular and traffic datasets has been limited. Popular examples such as KITTI [8], MOTChallenge 2015 [9], and UA-DETRAC [10] present benchmark computer vision tasks that are intended for autonomous vehicles and object tracking. But none of these datasets are intended to use a camera network structure to gain traffic insights. The TRANCOS dataset [11] does use a camera network in the highway system surrounding Madrid, Spain. However, the purpose of this work is only to provide a benchmark for counting vehicles in scenes with high vehicle overlap. Zhang et al. introduce the WebCamT dataset [12] from a web camera network in New York City in order to understand traffic density and create another vehicle counting benchmark alongside TRANCOS. Again, there is no aggregation of traffic data or use of the camera network structure to gain community-wide insights. Furthermore, all of these datasets, with the exception of KITTI, restrict themselves to highways and metropolitan areas. Suburban traffic behavior is not captured.

There are many other traffic camera datasets that utilize fewer than ten cameras and limit analysis to one intersection at a time [13, 14, 15, 16, 17, 18, 19, 20]. We refer interested readers to [21] for a more detailed survey of these datasets and related challenges. To the best of our knowledge, STREETS is the first vision-based dataset to temporally accumulate traffic data, utilize the inherent graph structure of the camera network, and capture the suburban traffic setting. And while many cities and transportation agencies provide publicly accessible web camera data, we hope STREETS is a meaningful step towards processing and curating these data sources for the research community.

In addition to vision-based datasets, many cities and states make in-roadway sensor data available to monitor traffic volumes and velocities. Perhaps the most popular such resource is the Performance Measurement System (PeMS) data source from the California Department of Transportation [22]. PeMS has been used as a benchmark for network-based traffic forecasting in recent work such as [23] and [24], for example. Again, we would like to point out that this data source only examines the California highway system around major metropolitan areas like Los Angeles and San Francisco. We argue that the suburban setting offers unique challenges with the *gated* nature of traffic flow from traffic signals and the common loss of *traffic mass* to residential and business locations. Furthermore, previous work has used vehicle speed data from PeMS while we present traffic volume data via counting vehicles in still images.

Lastly, few datasets exist to document non-recurring incidents that affect traffic conditions like car accidents or on-road debris. Chan et al. [25] extract on-vehicle dashcam footage from YouTube to anticipate car accidents, but this context is certainly different from ours with third-person view and a collection of traffic cameras. Shah et al. [26] do collect annotated traffic camera data for accident forecasting; however, their analysis is limited to prediction at a single view. We believe the incident data we provide along with our traffic graphs may lead to interesting work in estimating and identifying the impact of non-recurring incidents in a traffic graph.

## 2.2 Traffic Forecasting and Network Science

We would also like to describe previous work in the domain of traffic forecasting. Much attention has been given recently to how various modes of deep learning and graph neural networks (GNNs) may be used for predicting traffic volumes or speeds [23, 24, 27, 28, 29]. The work in [23, 24, 27] all use the aforementioned PeMS data source [22], while [28] and [29] use in-roadway sensors around Beijing and Seattle, respectively. These methods, while effective in utilizing the structure of a traffic graph, have been applied to data with narrow context and suffer from the lack of interpretability that is inherent in deep learning. There are of course simpler, classical baseline models for traffic forecasting and we refer the interested reader to [30] to learn more about these methods.

Traffic graphs have also attracted interest from researchers in the emerging area of Graph Signal Processing (GSP). Graph wavelets have been used to classify nodes in a traffic network [31], infer mobility patterns [32], and detect anomalous events [33]. Popular datasets in graph signal processing include the Minnesota highway road network [34] and a taxi-ride dataset in New York city [35]. Work like [36] typically uses the Minnesota road network by placing synthetic data on the graph and [37] details how to convert taxi-ride data into a coherent graph structure. For an overview of the current state of GSP, we refer to the excellent survey provided by Ortega et al. [38]

Lastly, computer vision has been explored as a mode of traffic analysis. Existing work typically examines vehicle detection and counting or traffic congestion at individual cameras [12, 39, 40, 41, 42, 43]. Less work has been done to combine observations from multiple cameras. Choe et al. attempt to match vehicles in a low frame-rate camera network as a proxy for estimating traffic congestion [44], but more work can certainly be done in this area. It is clear that traffic forecasting is appealing to researchers from multiple disciplines and backgrounds. We hope the variety of data sources in STREETS such as large-scale image data, data-efficient traffic metrics, and the graph-based representation motivate work within and between these respective fields.

# 3  STREETS

We now turn our attention to introducing our dataset: STREETS. We will discuss in detail the publicly-available web cameras we accessed for this work, the computer vision algorithm used to count vehicles, and the directed graphs we use to depict the camera network.

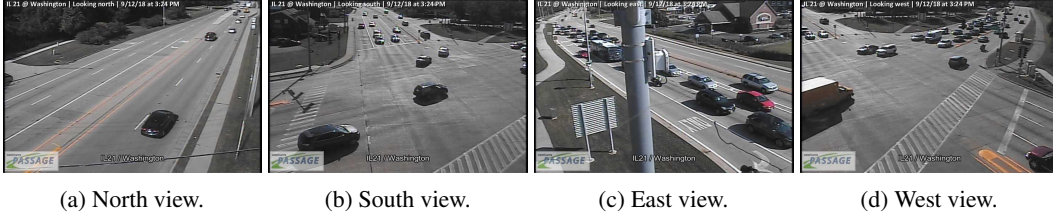

| (a) North view. | (b) South view. | (c) East view. | (d) West view. |

Figure 1: Example traffic camera images at each directional view for one camera location. Each image stream is available from its own publicly-accessible URL.

## 3.1 Data Collection

Lake County, IL is the suburban county north of Chicago, IL with a population of 700,000. Lake County PASSAGE (LCP) is the local transportation agency that maintains the publicly-available web camera network across the county. The objective of LCP is to provide up-to-date roadway warnings and traffic information for local law enforcement, news authorities, and the public at-large. Each camera operated by LCP is a pan-tilt-zoom (PTZ) camera located at the corner of a traffic intersection. Each camera rotates in place to observe each direction (North, East, etc.) of traffic at the intersection. Thus, each camera is capable of observing at least two and no more than four views at its location. The cameras do stream video data back to the LCP headquarters; however, due to bandwidth and storage limitations, publicly-available images are captured approximately every 10 minutes. At capture time, a camera rotates and saves an image at each view when stationary and the resulting images are stamped with the location and time at the top of each image before being published online for the public. Each camera location only takes a few seconds to collect images from all of its directional views. Thus, when we refer to a "camera location capture", we are referring to the act of one camera publishing images of each of its directional views. Figure 1 displays an example of a camera location capture. We observe that the views at just one camera location are capable of demonstrating diverse challenges in perspective, lighting, and sources of occlusion.

It is important to note that the camera locations capture images *asynchronously* with respect to one another and *aperiodically* with respect to themselves. To be clear, each distinct camera location capture occurs at different times, not simultaneously, and within a single location the time between captures is not consistent. Again, the time between captures is typically around 10 minutes. Of course, this low level of temporal resolution is unfortunate. To understand complex time-series behavior, we would like a higher temporal sampling rate. However, we note that the limitations for LCP in bandwidth and storage are a common constraint for any transportation agency, especially considering that Lake County is the wealthiest county in Illinois [45]. Not all agencies have the funding or scope to operate at the economies of scale that can enable more favorable data curation. While challenging, we encourage researchers to take up the task of developing intelligent transportation systems from infrastructure that is economical for communities of all sizes and means.

At the time of data collection for this work, LCP provided 351 distinct cameras across Lake County for a total of 1,142 different directional views of traffic. The cameras have varying levels of resolution, though the typical camera is of "standard definition" (like in Fig. 1) while some other cameras provide higher resolution. For the purposes of STREETS, we select two communities in Lake County of 50 camera locations each for a total of 100 distinct cameras. We collected images in two disjoint periods in the summers of 2018 and 2019. First, we extracted images every ten minutes from 5am to 11pm from 8/21/2018 to 9/20/2018 across the 320 views of the 100 camera locations. Second, we collected images every five minutes from 5am to 11pm from 6/5/2019-7/18/2019 at the same cameras. It is important we note that the images from 2019 were sampled twice as fast due to small improvements in image publishing rate from LCP. All told, STREETS offers over 4 millions still images across 2.5 months. Thus, the provided image data presents great variety in camera perspective, lighting conditions, occlusion sources, weather conditions, etc.

## 3.2 Traffic Data

We capture the traffic state by counting cars in each image. We use the Mask Region-based Convolutional Neural Network (Mask R-CNN) [46] as our vehicle detector to count cars in each frame.

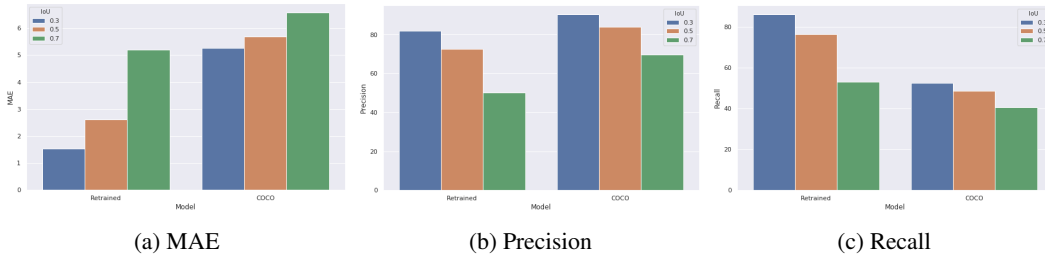

(a) MAE                    (b) Precision                    (c) Recall

Figure 2: Detector benchmarks on validation data with three choices of Intersection over Union (IoU) for positive match. Validation set contains 472 images for a total of 5,218 labeled vehicles.

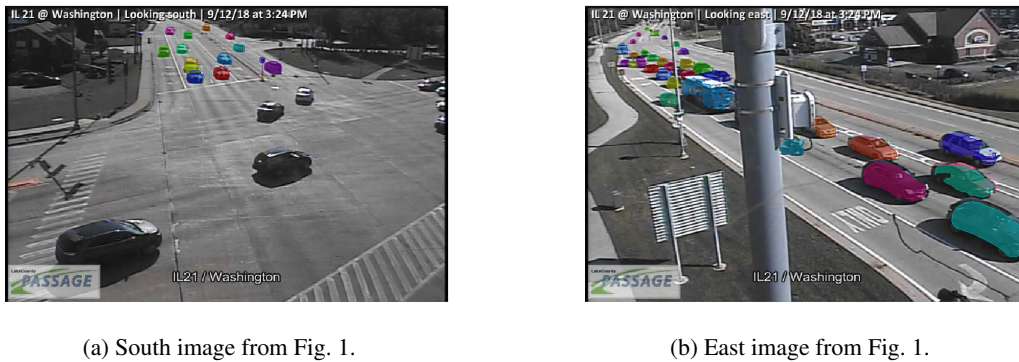

(a) South image from Fig. 1.                    (b) East image from Fig. 1.

Figure 3: Detection examples for our retrained Mask R-CNN vehicle detector. Background and intersection vehicles are excluded and the minimum detection confidence for each vehicle is 0.7.

The Mask R-CNN provides semantic segmentation for an input image and is derived from the Faster R-CNN [47]. To adapt the Mask R-CNN to our application of suburban traffic cameras, we retrain the network on a collection of nearly 3,000 labeled images with an 85-15 training-validation split. The images are sampled evenly across all camera views and nine different days during the month-long period in 2018. Greater emphasis is placed on images during rush hour periods (7-9am and 4-6pm) to improve the Mask R-CNN's ability to count vehicles in crowded scenes. We use the Tensorflow implementation of the Mask-RCNN from Matterport [48] and retrain starting from the provided COCO challenge network weights. Each labeled image has segmentation masks for each distinct, visible vehicle and we only consider vehicles on the main roadway of the current camera view. This means we exclude *background vehicles* in parking lots, driveways, etc. and do not count vehicles inside the intersection. In order to do so, we hand-label masks for the inbound (driving towards the camera) and outbound (driving away from the camera) sides of the roadway at each camera view. Across the 3,000 labeled images, we have 33,260 semantically segmented vehicles collected from workers on Amazon Mechanical Turk. Figure 2 provides vehicle counting metrics using the original COCO network weights from [48] against the retrained network weights from our training set of 2477 images with 28,042 vehicles. We use a detection confidence threshold of 0.7 and perform tests over three choices of Intersection over Union (IoU). The metrics we present are Mean Absolute Error (MAE) in vehicle counting per image, precision across the validation set, and recall across the validation set. Figure 3 displays examples of the retrained Mask R-CNN labeling images. For each image, we collect separate vehicle counts on the inbound and outbound sides of the roadway, respectively: every camera view in STREETS depicts a two-way roadway. We extract the timestamp from each image using the open-source Tesseract OCR software for more precise temporal alignment.

We also received documented incident data from LCP that coincides with the period of our data collection in 2018. These incidents are any non-recurring event that may impact traffic like car accidents or roadway debris. Each incident has an associated location, cause, and estimated traffic impact according to the traffic engineers at LCP. The incidents are intended to be documented as close to the time of the event as possible. Thus, the potential traffic impact should be properly aligned with the documented time. After filtering this data for incidents that are within 2 km of any of our

| Level | 1 | 2 | 3 | 4 |
|---|---|---|---|---|
| Count | 1440 | 543 | 565 | 18 |

Table 1: Distribution of traffic impact levels. Level 1 is lowest and Level 4 is highest.

| Type | Accident | Debris | Stall | Other |
|---|---|---|---|---|
| Count | 1018 | 462 | 453 | 633 |

Table 2: Distribution of incident types. Other contains less frequent types like fire or roadwork.

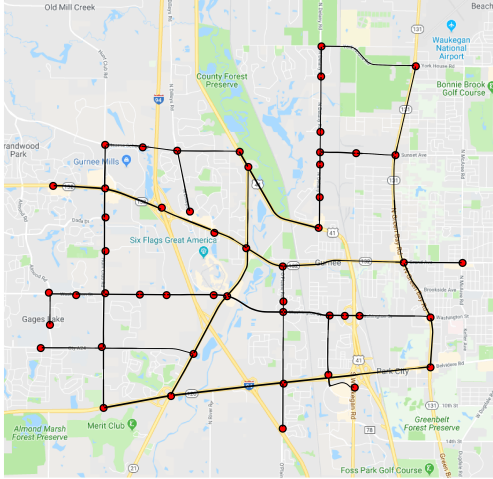

(a) Graph of Gurnee community.

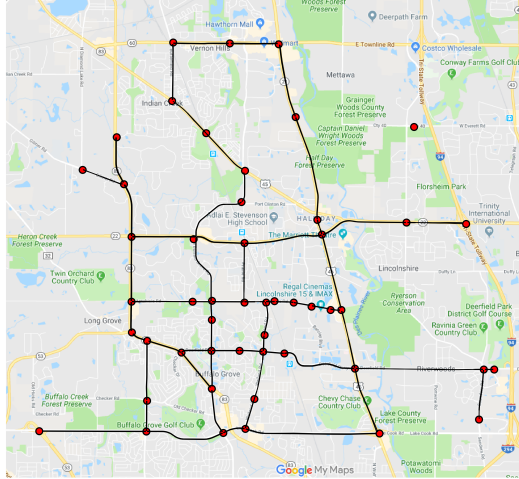

(b) Graph of Buffalo Grove community.

Figure 4: Graph structure of each community. For interpretability, we only show edges between camera locations. Each depicted edge is reciprocated in each direciton.

100 camera locations, we have a total of 2,566 non-recurring incidents in STREETS. Tables 1 and 2 illustrate the distribution of these incidents.

## 3.3 Camera Network Graph

We separate the 100 traffic cameras into two communities of 50 camera locations each. The two communities are centered about the towns of Gurnee, IL and Buffalo Grove, IL. For each community, we construct a directed graph to describe the underlying camera and road networks. We separate the inbound and outbound sides of each camera view into separate vertices in the directed graph. Thus, for a community with $N$ camera views we have $2N$ vertices. There are two types of edges in each directed graph: *internal edges* and *external edges*. For an edge to exist between two vertices, there must be at least one visible lane of traffic at both the source and destination vertices that may convey traffic. An internal edge connects an inbound camera view to an outbound camera view at a single location. For example, a left-turn lane permits a directed edge from an inbound North vertex to an outbound East vertex at the same camera location. An external edge connects an outbound edge of one location to the inbound edge of another location that is within 4 km. For example, there may be a directed edge between the outbound South vertex of one location to the inbound North vertex of a location to its South. The typical distance between two camera locations is around 1 to 2 km. We provide the number of traffic lanes and the Google Maps travel distance connecting two vertices as candidates for describing edge weights. Figures 4 and 5 visualize the graph structure of the two communities and one intersection in detail, respectively. Table 3 provides statistics regarding the scope of each community.

|  | Gurnee | Buffalo Grove |
|---|---|---|
| # Vertices | 318 | 322 |
| # Edges | 455 | 482 |
| Area (sq. km) | 81.55 | 110.62 |

Table 3: Statistics for each community graph.

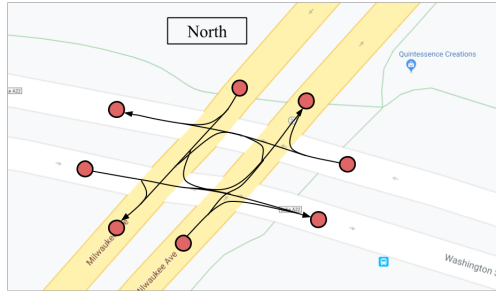

Figure 5: Internal edges for intersection in Fig. 1. There are no "u-turn" edges.

## 4 Benchmark Task and Discussion

The diverse data sources in STREETS may motivate a great variety of benchmarking tasks. Though we do not have the space to present many of these tasks in detail, we will explore one such task: single-step traffic prediction. We will list other interesting tasks in the concluding Section V.

For a graph with vertices $V$, we seek to predict the number of vehicles at a target vertex $v \in V$ at a future time index. To better understand the spatio-temporal dynamics of this prediction problem, we utilize traffic count data in the $K$-hop neighborhood of $v$ and use $L$ current/previous samples at each neighbor. Since the cameras from LCP operate asynchronously and aperiodically, we must choose a resampling and interpolation strategy to temporally align the data at each vertex. For our experiments, we fix the sampling rate to be 5 minutes and use a nearest-neighbor interpolation strategy.

### 4.1 Traffic State Inference

The gated nature of the road network in STREETS presents a distinct challenge. Consecutive samples at a given camera can give very different impressions of traffic depending on the state of the traffic intersection. For example, during rush hour, we may see twenty cars waiting at an intersection in one image, then two in the next image if we are witnessing the end of the signal clearing the traffic queue. Figure 6 demonstrates how such large variations can occur in adjacent samples

As such, we propose the use of a simple binary attribute to indicate whether a sample from a camera is in a *queue* state (Q) or a *free-flow* (F) state. For example, the images in Fig. 1 would be labeled as F, Q, Q, Q, from left to right. The intuition here is to provide better context for each sample and condition a model's understanding of the data available to it. To accomplish this, we labeled 3,500 images as being in either queue or free-flow state for the roadway of interest and trained a ResNet-50 [49] model with an 80-20 training-validation split to perform the binary classification. In brief, this model achieves 99% training and 86% validation accuracy, which is suitably effective given that the distinction between queue and free-flow is not completely objective. Codes and data for this model are available on our accompanying GitHub[2] and dataset, respectively.

### 4.2 Baseline Methods

We compare five baseline methods for the single-step traffic prediction problem. The first is a simple historical average. For a given vertex $v$ and time index $n$, the historical average (HA) is the sample mean of all samples collected at that vertex and time index. The remaining methods utilize feature vectors constructed as the concatenation of: the present time index $n$ and the last $L$ samples for each vertex in the neighborhood. Here, a sample refers to both the number of vehicles and the predicted traffic state of the corresponding image as a binary feature. The ground truth is the vehicle count at vertex $v$ at index $n + \delta$, where we predict $\delta$ samples in the future. The other models we test are Random Forest Regressor (RFR), Support Vector Regressor (SVR), Linear Regression (LR), and a simple artifical neural network (ANN) with two hidden layers.

| Model | $L$ | $K$ | 7/15 (MAE$_c$/MAE) | 7/16 | 7/17 | 7/18 |
|---|---|---|---|---|---|---|
| RFR | 3 | 0 | 4.21/4.44 | 3.44/4.05 | 3.40/4.21 | 3.80/4.19 |
| | | 2 | **3.85**/4.10 | 3.09/3.41 | 3.28/4.19 | 3.52/3.64 |
| | 6 | 0 | 4.14/4.31 | 3.28/3.42 | 3.43/4.20 | 3.63/4.30 |
| | | 2 | 4.11/4.27 | 3.03/3.27 | 3.28/4.17 | 3.54/3.77 |
| SVR | 3 | 0 | 4.07/4.28 | 3.05/3.25 | 3.21/3.90 | 3.56/3.83 |
| | | 2 | 4.00/4.24 | **2.96**/3.20 | **3.10**/3.92 | **3.44**/3.74 |
| | 6 | 0 | 4.16/4.29 | 3.14/3.32 | 3.25/3.92 | 3.54/3.81 |
| | | 2 | 4.15/4.31 | 3.09/3.33 | 3.13/4.03 | 3.50/3.70 |
| LR | 3 | 0 | 4.80/5.05 | 3.53/3.79 | 4.08/4.79 | 3.95/4.49 |
| | | 2 | 4.42/4.55 | 3.10/3.39 | 3.53/4.46 | 3.45/3.92 |
| | 6 | 0 | 4.95/4.95 | 3.69/3.97 | 4.09/4.94 | 3.92/4.41 |
| | | 2 | 4.57/4.61 | 3.41/3.55 | 3.54/4.64 | 3.61/4.04 |
| ANN | 3 | 0 | 4.23/4.45 | 3.21/3.40 | 3.46/4.23 | 3.82/3.99 |
| | | 2 | 4.44/4.62 | 3.52/3.73 | 3.62/4.40 | 3.81/4.24 |
| | 6 | 0 | 4.60/4.63 | 3.32/3.42 | 3.42/4.29 | 3.75/3.90 |
| | | 2 | 5.16/5.30 | 3.49/3.66 | 3.72/4.64 | 4.02/4.32 |
| HA | | | 4.03/4.16 | 3.11/3.17 | 3.20/4.05 | 3.64/3.77 |

Table 4: Prediction results for 15 minute horizon. The best result for each date is bolded.

## 4.3  Experimental Setup and Results

We conduct experiments at the camera view depicted in Fig. 1.(c). We predict the vehicle counts on the inbound vertex for each date in 7/15/2019-7/18/2019 and vary the choices of $K$, $L$, and $\delta$ to examine the importance of the traffic graph. Furthermore, we compare the effects of conditioning predictions on the traffic state of the future sample. We use MAE as our performance metric and thus the unconditioned and conditioned metrics are given by the following. Let $N$ be the number of samples we predict, $\{y_i\}_{i=1}^{N}$ the ground truth traffic data, $\{\hat{y}_i\}_{i=1}^{N}$ the unconditioned predictions, and $\{\hat{y}_{i|\text{state}}\}_{i=1}^{N}$ the conditioned predictions:

$$\text{MAE} = \frac{1}{N} \sum_{i=1}^{N} |y_i - \hat{y}_i| \tag{1}$$

$$\text{MAE}_c = \frac{1}{N} \sum_{i=1}^{N} |y_i - \hat{y}_{i|\text{F}}| \mathbb{1}\{y_i \in \text{F}\} + |y_i - \hat{y}_{i|\text{Q}}| \mathbb{1}\{y_i \in \text{Q}\} \tag{2}$$

The RFR, SVR, and LR models are all constructed using their corresponding default models in the scikit-learn Python library. The ANN model is implemented in PyTorch with hidden layers of sizes 200 and 100. Training is performed for 10,000 steps with batch size of 32 using Stochastic Gradient Descent at learning rate $10^{-3}$ and momentum 0.9. Traffic data from the weekdays of 6/5/2019-7/12/2019 are used as training data for each model. When MAE$_c$ is used, we split the training data according to the traffic state of the corresponding ground truth and train two parallel models for the F and Q data. For each experimental setting, we conduct 10 trials and take the mean error measure.

Tables 4 and 5 present results for each model with varying choices of $L$ and $K$ under both MAE and MAE$_c$. Overall, the non-linear methods of RFR and SVR (with radial basis function kernel by default) are consistently the top performing methods. Furthermore, we see that the larger 2-hop neighborhood provides better results for each method except for the ANN. This nicely confirms the value of the traffic graph in prediction. Conditioning the model predictions on the traffic state also consistently improves model performance as the MAE$_c$ values are almost always smaller than the MAE values. The poor results for our ANN method may be simply due to a lack of model search and tuning. Finally, we see an interesting phenomenon in the difference in errors between each day. For example, 7/16 and 7/17 have errors around 3.1, while 7/15 has errors around 4.1 even though they are all weekdays in the same week. Figure 6 shows the ground truth and SVR predictions for the 7/15 and 7/16 data to visualize how their samples differ. We see in both plots that the conditioned model using MAE$_c$ is more capable of tracking rapid changes in the traffic data while the unconditioned model lies in between as it effectively smooths the data.

| Model | $L$ | $K$ | 7/15 (MAE$_c$/MAE) | 7/16 | 7/17 | 7/18 |
|---|---|---|---|---|---|---|
| RFR | 3 | 0 | 4.07/4.30 | 3.30/3.44 | 3.45/4.27 | 3.86/4.26 |
| | | 2 | 4.10/4.22 | 3.06/3.32 | 3.26/4.18 | 3.65/4.18 |
| | 6 | 0 | 4.14/4.44 | 3.43/3.85 | 3.67/4.63 | 3.54/3.66 |
| | | 2 | **3.98**/4.06 | **2.99**/3.21 | 3.22/4.08 | 3.52/3.83 |
| SVR | 3 | 0 | 4.15/4.28 | 3.12/3.32 | 3.23/3.91 | 3.53/3.80 |
| | | 2 | 4.13/4.30 | 3.08/3.32 | 3.11/4.00 | **3.50**/3.70 |
| | 6 | 0 | 4.16/4.28 | 3.04/3.29 | 3.19/3.85 | 3.62/3.81 |
| | | 2 | 4.16/4.30 | 3.00/3.24 | **3.10**/3.93 | 3.49/3.80 |
| LR | 3 | 0 | 4.95/4.95 | 3.69/3.97 | 4.09/4.94 | 3.92/4.41 |
| | | 2 | 4.56/4.61 | 3.41/3.55 | 3.55/4.64 | 3.60/4.04 |
| | 6 | 0 | 4.85/4.92 | 3.60/3.90 | 4.10/4.80 | 4.02/4.31 |
| | | 2 | 4.31/4.53 | 3.32/3.61 | 3.67/4.49 | 3.62/4.04 |
| ANN | 3 | 0 | 4.65/4.57 | 3.29/3.44 | 3.43/4.27 | 3.75/3.88 |
| | | 2 | 4.82/4.92 | 3.64/3.59 | 3.63/4.54 | 3.87/4.03 |
| | 6 | 0 | 4.54/4.56 | 3.22/3.47 | 3.45/4.21 | 3.81/3.95 |
| | | 2 | 4.65/5.06 | 3.62/4.23 | 4.13/4.80 | 3.87/4.47 |
| HA | | | 4.13/4.22 | 3.22/3.35 | 3.24/3.96 | 3.74/4.01 |

Table 5: Prediction results for 30 minute horizon. The best result for each date is bolded.

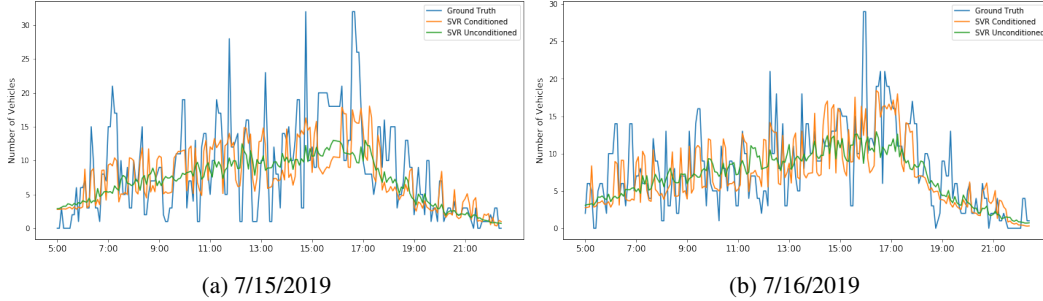

(a) 7/15/2019　　　　　　　　　　　　　　　(b) 7/16/2019

Figure 6: Example vehicle count predictions for 15 minute horizon with SVR ($K = 2$, $L = 3$) using MAE (unconditioned) and MAE$_c$ (conditioned)

## 5   Conclusion and Future Work

We have motivated the need for more diverse traffic flow datasets in both context and data source. We presented our dataset, STREETS, and explained the variety of available data including large-scale image data, non-recurring traffic incidents, and directed traffic graphs. Benchmarking results demonstrated the value of both temporal and spatial information from the traffic graph to predict future traffic states. We believe there are many other interesting tasks STREETS may motivate, but do not have the space to explore. Such tasks include traffic density estimation, non-recurring incident identification and traffic impact forecasting, vehicle counting, semantic annotation of scenes from web cameras, and background-foreground separation. We will consider updates and improvements to the dataset according to the needs of the research community. We want to make improvements that can facilitate practical solutions in intelligent transportation systems for communities of any size and economic means.

## Acknowledgements

This work was partly supported by a Lab Directed Research and Development grant from Sandia National Laboratories[3]. We would also like to thank Lake County Passage and its employees for their cooperation and answering our questions while performing this work.

## Footnotes

[1]Link to STREETS dataset. [7]

[2]Link to STREETS GitHub.

[3]This paper describes objective technical results and analysis. Any subjective views or opinions that might be expressed in the paper do not necessarily represent the views of the U.S. Department of Energy or the United States Government. SAND2019-11477 C.

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
