[Reviews · NeurIPS 2019]

Reviewer 1



In general, the article is very well written, although at certain places the readers will be easily lost because of some convoluted explanations of technical details, e.g.; - why edges are divided in two types: internal and external? - Are them used differently in benchmarking? - Are those non-recurring traffic-related incidents collected used in benchmarking? - What can they be used for and how? - etc. On the other hand, the paper is well introduced and motivated, raising very challenging problems which may be studied using STREETS. Also, it is very well connected with the literature: section two is a fine and complete summary of existing literature on traffic and vehicle datasets. However, experiments (benchmark evaluation) are not so convincing. The experimental section should be improved, as there is a clear lack of insight as to how “STREET” may be useful to validate and develop new ML models in the area, or how it could be used to guide future work using the dataset. Authors only show two simple baselines addressing one task, which is clearly insufficient to draw any conclusions. Much more tasks and SOTA baselines should have been included to demonstrate the potential of the dataset. Authors argue that this is due to the lack of space, however they could have reduced the description of the procedures and algorithms for data collection, or they could have used the appendix. ------------------- I am changing from "an OK submission, but not good enough" to "marginally above acceptance threshold" after reading the response an the rest of reviews. I appreciate the additional results provided by the authors.

Reviewer 2



This paper describes an original contribution of a large dataset of street traffic images. The quality of the dataset is excellent and the manuscript is written very clearly, with a great description of the background and prior work. Overall I find this dataset to be a very significant contribution to science.

Reviewer 3



### Edit: Dear authors, Thank you for your rebuttal. I encourage you to continue to expand the baselines, include the details about the annotations in the appendix of the paper, and work on providing a well-documented code release. I trust you will do all of these things. Though I'm impressed with the rebuttal, I will probably not change my score since I think it is already quite high. I do however think the paper should be accepted. Thanks for your contribution! Best, R3 ### Originality -STREETS differs from previous work in several ways: --The dataset is collected from a camera network with a graph-based structure, and the relationship between cameras is available --The dataset focuses on the suburban setting --The dataset accumulates temporal traffic data from multiple intersections Overall, the authors have clearly explained how STREETS is different from prior work and have explicitly developed STREETS to address shortcomings in this work. Quality -The process for collecting the dataset is straightforward, but clearly described, which contributes to the technical merit of the paper. -The authors contribute a rich amount of metadata, including: --Timestamp, location, view for each image --Hand-labeled masks for cars for the inbound and outbound sides of the roadway at each camera view --Number of traffic lanes at each camera, location of camera, and distance between cameras for recontstructing the traffic graph --Incident data from local transportation authorities -The authors evaluate state of the art object detection networks (Mask-RCNN) for counting vehicles in crowded scenes. -The authors include baseline evaluations on the STREETS dataset for the single-step traffic prediction problem Clarity -The submission is clearly written and well organized. I found that the authors did a particularly good job of motivating the problem and I enjoyed the discussion of the tradeoffs of various traffic forecasting data sources in the introduction. The related work was also particularly well written and informative. I am not an expert in traffic forecasting, but after reading the paper, I understood the problem the researchers were trying to address and had enough context to know why their contribution is significant. Significance -The STREETS dataset addresses many shortcomings found in prior work. Many more researchers in the NeurIPS community should be building datasets like this and I applaud the authors for their efforts. Additionally, many researchers will benefit from this dataset and the work will inspire the development of new methods that will help advance current technology for a meaningful problem.

[Author Response · NeurIPS 2019]

*General Response:*
We thank the reviewers for their comments and look forward to improving our work from their suggestions. Below we identify and address questions posed by the reviewers.

*Completeness of Benchmarking Tasks:* We agree that the evaluation of benchmarking tasks deserves more attention as R1 and R3 pointed out. We believe that having more training data and at a faster temporal sampling rate would improve our ability to test STREETS with existing and state-of-the-art (SOTA) methods for traffic prediction. Fortunately, the infrastructure for Lake County Passage (LCP) has been improved this summer and we were able to collect another 1.5 months of images at a temporal sampling rate of every five minutes as opposed to the original ten minutes. We plan to give more depth to the benchmarking of STREETS in the coming months to give better context to the reader.

*Permanent Storage of the Dataset:* Storing our dataset on Dropbox is a temporary solution to anonymize our submission. We will permanently store STREETS on a data bank associated with our university once we no longer need to remain anonymous. This data bank guarantees that our data will be publicly accessible to anyone in the world, have a stable DOI for citation, and be accessible for at least five years (further-extended storage is typical). In addition, the data bank provides a data curation staff to ensure datasets are well-documented, readable and easily found via Google Scholar. We have paraphrased these points to preserve the anonymity of the data bank's website. Furthermore, storing on this public data bank would place the dataset under a CC0 license as R2 suggested. We will also consult with the employees of the data bank to decide if the name STREETS would make locating our dataset more difficult for researchers. We are willing to change the name should it make the dataset more accessible as R2 noted.

*Code Repository:* We will also create a GitHub for STREETS to allow researchers to replicate our data generation procedure and also make handling our existing data easy and intuitive. This would include our code for counting vehicles, extracting image timestamps, visualizing parts of the camera network, loading individual camera streams, and more. These codes would be in Python; however, we will convert any existing Python-specific file types to a language-agnostic format like hdf5.

*Quality of Hand Annotated Vehicles:* We hand-reviewed every hand annotation of the cars from the workers on Amazon Mechanical Turk (MTurk). Workers were expected to tightly outline every individually identifiable vehicle. For example, highly overlapping vehicles at an intersection are individually labeled if they are distinguishable from one another. Conversely, small vehicles in the distance are only labeled if they are clearly one vehicle, e.g. distinct headlights, bumper. Blobs or low detail masses are not labeled since they may be more than one car. Furthermore, workers were told to only label vehicles on the "main roadway" for the given camera view. This excludes vehicles that are entirely inside the intersection, background vehicles in parking lots, and vehicles on other legs of the intersection. For example, if any cars are visible on the North leg while looking at the East leg, we ignore those "North" vehicles since they should be captured by the camera when it looks North. These constraints prevent overcounting and guarantee we know the source or destination road for each vehicle. We do not have a precise number on the labeling accuracy; however, we can say with high confidence that at least 90% of eligible vehicles are appropriately labeled. Figure 1 gives a couple examples of labeled images.

|          (a)          |          (b)          |

Figure 1: Example vehicle annotations from workers on MTurk. Each individual and distinguishable vehicles receives its own outline. In (a), every vehicle is identifiable and outlined while in (b) some distant vehicles and taillights make separating vehicles too difficult, thus these vehicles are left alone. Also note that background vehicles in parking lots are properly excluded from (b).

[Meta-Review · NeurIPS 2019]

The paper presents a new dataset with recordings and annotations of camera network for the analysis of traffic flow. The main originality is the graph based structure of the relationships between cameras. The Reviewers agree that the dataset has been properly designed and might have a potential impact on scientific community. The paper includes an empirical evaluation of some reference methods.